# Charge-4e superconductivity and chiral metal in 45°-twisted bilayer cuprates and related bilayers

Yu-Bo Liu[1,8], Jing Zhou[2,3,8], Congjun Wu [4,5,6,7] & Fan Yang [1]✉

The material realization of charge-4e/6e superconductivity (SC) is a big challenge. Here, we propose to realize charge-4e SC in maximally-twisted homobilayers, such as 45°-twisted bilayer cuprates and 30°-twisted bilayer graphene, referred to as twist-bilayer quasicrystals (TB-QC). When each monolayer hosts a pairing state with the largest pairing angular momentum, previous studies have found that the second-order interlayer Josephson coupling would drive chiral topological SC (TSC) in the TB-QC. Here we propose that, above the $T_c$ of the chiral TSC, either charge-4e SC or chiral metal can arise as vestigial phases, depending on the ordering of the total- and relative-pairing-phase fields of the two layers. Based on a thorough symmetry analysis to get the low-energy effective Hamiltonian, we conduct a combined renormalization-group and Monte-Carlo study and obtain the phase diagram, which includes the charge-4e SC and chiral metal phases.

The charge-4e/6e superconductivities (SCs) are exotic SCs characterized by $\frac{1}{2}/\frac{1}{3}$ flux quantization. These novel SCs are formed by condensation of electron quartets/sextets[1-25], which is beyond the conventional Bardeen–Cooper–Schrieffer mechanism[26]. Recently, it was proposed that these intriguing SCs can emerge as the high-temperature vestigial phases of the charge-2e SC in systems hosting multiple coexisting pairing order parameters (ODPs). Typical proposals for such multi-component pairings include the incommensurate pair-density-wave (PDW)[10,11,14], the nematic pairing[17,18], and the bilayer pairing system[16,19]. However, each proposal is still waiting for the experiment's realization.

One proposal is through melting of incommensurate PDW[10,11,14]. The PDW has been reported in such materials as the cuprates[27,28], the $CsV_3Sb_5$[29], and the transition-metal dichalcogenide[30]. This proposal, however, suffers from the difficulty that the PDWs observed in these experiments are always accompanied by a dominant uniform SC part. Another proposal is through the melting of nematic pairing[17,18]. Such a

pairing state is formed through the real mixing of the two basis functions of a two-dimensional (2D) irreducible representation (IRRP) of the point group. More recently, a group-theory based classification of the vestigial phases generated by melting of the pairing states belonging to the 2D IRRPs was performed[31], wherein such interesting phase as $d$-wave charge-4e SC was proposed. However, the experiment verification of these proposals is still on the way. Alternatively, a bilayer approach was recently proposed[16,19] in which, two monolayers hosting SCs with different phase stiffness are coupled. Consequently, in an intermediate-temperature vestigial phase, one layer carries charge-2e SC while the other layer carries charge-4e SC[19]. The drawback of this proposal lies in that, in an out-of-plane magnetic field, while the charge-4e-SC layer allows for integer times of half magnetic flux, the charge-2e-SC layer only allows for integer flux. As the two layers experience the same magnetic flux, only the integer flux is allowed, and the hallmark of the charge-4e SC, i.e., the half flux quantization, cannot be experimentally detected in this proposal. Finally, the melting of the

[1]School of Physics, Beijing Institute of Technology, Beijing 100081, China. [2]Department of Science, Chongqing University of Posts and Telecommunications, Chongqing 400065, China. [3]Institute for Advanced Sciences, Chongqing University of Posts and Telecommunications, Chongqing 400065, China. [4]Institute for Theoretical Sciences, WestLake University, 310024 Hangzhou, China. [5]New Cornerstone Science Laboratory, Department of Physics, School of Science, Westlake University, 310024 Hangzhou, China. [6]Key Laboratory for Quantum Materials of Zhejiang Province, Department of Physics, School of Science, Westlake University, Hangzhou 310030, P. R. China. [7]Institute of Natural Sciences, Westlake Institute for Advanced Study, Hangzhou 310024, P. R. China. [8]These authors contributed equally: Yu-Bo Liu, Jing Zhou. ✉e-mail: yangfan_blg@bit.edu.cn

multi-component hexatic chiral superconductor leading to vestigial charge-6e SC was proposed in the context of kagome superconductors[22]. Presently, the material realization of the charge-4e/6e SC is still a big challenge.

Here in this work, we take advantage of the rapid development of the twistronics[32–61] and utilize it to design the intriguing charge-4e SC. Here we shall study materials made through stacking two identical monolayers with the largest twist angle, which host Moireless quasicrystal (QC) structures[62–64] and are dubbed as the twist-bilayer QC (TB-QC)[65], exampled by the recently synthesized 30°-twisted bilayer graphene[66–70] and 45°-twisted bilayer cuprates[71,72]. Prominently, the TB-QC hosts a doubly enlarged fold of rotation axis relative to its monolayer. Previous study[65,73] suggests that when each monolayer hosts a pairing state carrying the largest pairing angular momentum for the lattice, the second-order interlayer Josephson coupling (IJC) between the pairing ODPs from the two layers in the TB-QC makes them mix as $1: \pm i$, leading to time-reversal symmetry (TRS) breaking chiral topological SC (TSC). For example, as the monolayer cuprate carries the d-wave pairing, the 45°-twisted bilayer cuprates will host the $d+id$ chiral TSC[65,74–77]. It's interesting to investigate possible vestigial secondary orders above the $T_c$ of these chiral TSC phases, driven by the second-order IJC between the pairing ODPs from the two layers.

In this paper, we study the secondary orders in the superconducting TB-QC. Its unique symmetry leads to a simplified low-energy effective Hamiltonian including decoupled total- and relative-phase fields between the bilayer. Significantly, the second-order IJC allows the relative phase to fluctuate between its two saddle points to restore the TRS. Consequently, while the unilateral order of the relative-phase field leads to the TRS-breaking chiral-metal phase, the unilateral quasi-order of the total-phase field leads to the charge-4e SC phase, in which two Cooper pairs from different layers pair to form quartets. These two vestigial phases occupy different regimes in the phase diagram obtained by our combined renormalization group (RG) and Monte-Carlo (MC) studies, which are unambiguously identified by various temperature-dependent quantities including the specific heat, the secondary ODPs, and their susceptibilities, as well as the spatial-dependent correlation functions.

## Results

### Model and symmetry

Taking two $D_n$-symmetric monolayers, let's stack them by the twist angle $\pi/n$ to form a TB-QC, as shown in Fig. 1 for $n = 4$ (e.g., the cuprates) and $n = 6$ (e.g., the graphene). Obviously, the point group is $D_{nd}$, isomorphic to $D_{2n}$. There is an additional symmetry generator in the

TB-QC which is absent in its monolayer, i.e., the $C_{2n}^1$ rotation accompanied by a succeeding layer exchange, renamed as $\tilde{C}_{2n}^1$ here.

Suppose that driven by some pairing mechanism, the monolayer $\mu = t/b$ (top/bottom) can host a pairing state with pairing angular momentum $L = n/2$. While the cuprate monolayer hosting the d-wave SC synthesized recently[78] provides a good example for $n = 4$, some members in the graphene family which were predicted to host the f-wave SC[73,79,80] set an example for $n = 6$. The pairing gap function in the $\mu$ layer is

$$\Delta^{(\mu)}(\mathbf{k}) = \psi_\mu \Gamma^{(\mu)}(\mathbf{k}). \tag{1}$$

Here $\Gamma^{(\mu)}(\mathbf{k})$ is the normalized real form factor, and $\psi_\mu$ is the "complex pairing amplitude". Prominently, the $\Gamma^{(\mu)}(\mathbf{k})$ for $L = n/2$ changes sign with every $C_n^1$ rotation. As shown in Fig. 1, we choose a gauge so that

$$\Gamma^{(b)}(\mathbf{k}) = \hat{P}_{\frac{\pi}{n}} \Gamma^{(t)}(\mathbf{k}), \quad \hat{P}_{\frac{2\pi}{n}} \Gamma^{(\mu)}(\mathbf{k}) = -\Gamma^{(\mu)}(\mathbf{k}). \tag{2}$$

Here $\hat{P}_\phi$ indicates the rotation by the angle $\phi$. As the interlayer coupling in the TB-QC is weak[62,63,65], we can only consider the dominant intralayer pairing, but the two intralayer pairing ODPs can couple through the IJC[73–77,80]. We shall investigate the ground state and the vestigial secondary orders induced by this IJC.

Firstly, let's make a saddle-point analysis for the Ginzburg–Landau (G–L) free energy $F$ as functional of $\psi_{t/b}$. For the saddle-point solution, the $\psi_{t/b}$ are spatially uniform constant numbers. $F$ is decomposed as,

$$F(\psi_t, \psi_b) = F_0\left(|\psi_t|^2\right) + F_0\left(|\psi_b|^2\right) + F_J(\psi_t, \psi_b), \tag{3}$$

where $F_0(|\psi_\mu|^2)$ are the monolayers terms and $F_J$ is the IJC. The TRS-allowed first-order IJC takes the form,

$$F_J^{(1)}(\psi_t, \psi_b) = -\alpha(\psi_t \psi_b^* + c.c). \tag{4}$$

Under $\tilde{C}_{2n}^1$, the gap function on the $\mu$ layer changes from $\Delta^{(\mu)}(\mathbf{k}) = \psi_\mu \Gamma^{(\mu)}(\mathbf{k})$ to $\tilde{\Delta}^{(\mu)}(\mathbf{k}) = \psi_{\bar\mu} \hat{P}_{\frac{\pi}{n}} \Gamma^{(\bar\mu)}(\mathbf{k})$ which, under Eq. (2), can be rewritten as $\tilde{\psi}_\mu \Gamma^{(\mu)}(\mathbf{k})$ with

$$\tilde{\psi}_b = \psi_t, \quad \tilde{\psi}_t = -\psi_b. \tag{5}$$

The invariance of $F$ under $\tilde{C}_{2n}^1$ requires $\alpha = 0$. Thus, the following second-order IJC should be considered,

$$F_J(\psi_t, \psi_b) = A_0\left(\psi_t^2 \psi_b^{2*} + c.c.\right) + O\left(\psi^6\right). \tag{6}$$

Equation (6) is minimized at $\psi_b = \pm i\psi_t$ for $A_0 > 0$ or $\psi_b = \pm\psi_t$ for $A_0 < 0$. Previous microscopic calculations favor the former for the 45°-twisted bilayer cuprates[65,74] and 30°-twisted bilayer of the graphene family[73,80], leading to $d+id$ or $f+if$ chiral TSCs ground state.

Secondly, let us provide the low-energy effective Hamiltonian for the pairing-phase fluctuations. In this study, we fix $\Gamma^{(\mu)}$ and set $\psi_\mu \to \psi_\mu(\mathbf{r})$ as a slowly varying "envelope" function to describe the spatial fluctuation of the complex pairing amplitude. Focusing on the phase fluctuation, $\psi_{t/b}$ are written as $\psi_{t/b} = \psi_0 e^{i\theta_{t/b}(\mathbf{r})}$ where $\psi_0 > 0$ is a constant. The $\theta_{t/b}(\mathbf{r})$ are further written as

$$\theta_t(\mathbf{r}) = \theta_+(\mathbf{r}) + \theta_-(\mathbf{r}), \quad \theta_b(\mathbf{r}) = \theta_+(\mathbf{r}) - \theta_-(\mathbf{r}). \tag{7}$$

Here $\theta_+(\mathbf{r})$ and $\theta_-(\mathbf{r})$ denote the total and relative pairing phases. The low-energy effective Hamiltonian reads

$$H = H_0\left[\partial_\pm \theta_+, \partial_\pm \theta_-\right] + A_0\psi_0^4 \int \cos 4\theta_-(\mathbf{r}) d^2\mathbf{r}, \tag{8}$$

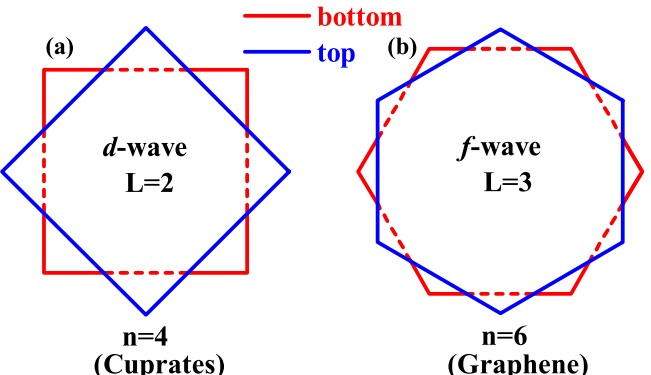

**Fig. 1 | Schematic illustration of a TB-QC formed by two $D_n$-symmetric monolayers, with each monolayer carrying SC with pairing angular momentum $L = \frac{n}{2}$.** As examples, $n = 4$ (cuprates) and $n = 6$ (graphene) are shown in (a) and (b), respectively.

with $\partial_\pm \equiv \partial_x \pm i\partial_y$. Up to the lowest-order expansion, the $H_0$ takes the following explicit form in the **k**-space,

$$H_0 = \frac{1}{2}\int d^2\mathbf{k}\Big[\theta_+(\mathbf{k})\theta_+(-\mathbf{k})\big(\alpha k_+^2 + \beta k_-^2 + \rho k_+ k_-\big) \\ + \theta_+(\mathbf{k})\theta_-(-\mathbf{k})\big(\omega k_+^2 + \delta k_-^2 + \eta k_+ k_-\big) \\ + \theta_-(\mathbf{k})\theta_-(-\mathbf{k})\big(\epsilon k_+^2 + \xi k_-^2 + \kappa k_+ k_-\big)\Big]. \tag{9}$$

Under $\tilde{C}_{2n}^1$, the gap function on the $\mu$ layer changes from $\Delta^{(\mu)} = \psi_\mu \Gamma^{(\mu)}$ to $\tilde{\Delta}^{(\mu)} = \psi_{\tilde{\mu}}\hat{P}_{\frac{\pi}{n}}\Gamma^{(\mu)}$ which, under Eq. (2), can be rewritten as $\tilde{\psi}_\mu \Gamma^{(\mu)}$ with

$$\tilde{\psi}_b(\mathbf{r}) = \psi_t\left(\hat{P}_{\frac{\pi}{n}}^{-1}\mathbf{r}\right),\ \tilde{\psi}_t(\mathbf{r}) = -\psi_b\left(\hat{P}_{\frac{\pi}{n}}^{-1}\mathbf{r}\right). \tag{10}$$

Consequently, we have

$$\theta_+(\mathbf{k}) \to \theta_+\left(\hat{P}_{\frac{\pi}{n}}^{-1}\mathbf{k}\right), \theta_-(\mathbf{k}) \to -\theta_-\left(\hat{P}_{\frac{\pi}{n}}^{-1}\mathbf{k}\right). \tag{11}$$

The invariance of Eq. (9) under (11) only allows for nonzero $\rho$ and $\kappa$, leading to the real-space Hamiltonian

$$H = \int d^2\mathbf{r}\left(\frac{\rho}{2}|\nabla\theta_+|^2 + \frac{\kappa}{2}|\nabla\theta_-|^2 + A_0\psi_0^4\cos 4\theta_-\right). \tag{12}$$

Equation (12) shows two important features. Firstly, the $\theta_+$ and $\theta_-$ fields are dynamically decoupled, with each hosting different stiffness parameter $\rho$ or $\kappa$ derived by the G-L expansion in the Sec. I of Supplementary Information (SI). Secondly, the second-order IJC allows $\theta_t - \theta_b = 2\theta_-$ to fluctuate between its two saddle points, i.e., $\pm\pi/2$, to restore the TRS. Note that although the term $\cos(4\theta_-)$ in Eq. (12) leads to four different values of $\theta_-$: $\pm\pi/4$ and $\pm 3\pi/4$ for the ground state, $\theta_- = \pi/4$ ($-\pi/4$) leads to gauge equivalent state with $\theta_- = -3\pi/4$ ($3\pi/4$). So the system only possesses two-fold Ising anisotropy. Here the unilateral quasi-ordering of the $\theta_+$ field leads to the ODP $\Delta^{(t)}(\mathbf{k})\cdot\Delta^{(b)}(-\mathbf{k})$ characterizing the charge-4e SC in which two Cooper pairs from different layers pair. The unilateral ordering of the $\theta_-$ field leads to the ODP $\Delta^{(t)*}(\mathbf{k})\cdot\Delta^{(b)}(\mathbf{k})$ characterizing the TRS breaking chiral metal[81,82]. Note that while $\theta_+$ and $\theta_-$ each can host either integer or half-integer vortices, Eq. (7) requires that they can only simultaneously host integer or half-integer vortices to ensure the single-valuedness of $\psi_{t/b}$[10,18]. This sets the "kinematic constraint" in the low-energy "classical Hilbert space" for allowed vortices of the two fields.

*RG study*: To perform the RG study, we start with the following effective action at the temperature $T$,

$$S = \int d^2\mathbf{r}\left(\frac{\rho}{2T}|\nabla\theta_+|^2 + \frac{\kappa}{2T}|\nabla\theta_-|^2 + g_4\cos 4\theta_-\right) \tag{13}$$

Here $g_4 > 0$ is proportional to $A_0$. This action can be mapped to a two-component Sine-Gordon model,

$$S_{SG} = \int d^2\mathbf{x}\left(\frac{T}{2\rho}\left|\nabla\tilde{\theta}_+\right|^2 + \frac{T}{2\kappa}\left|\nabla\tilde{\theta}_-\right|^2 + g_4\cos 4\theta_- - g_{2,0}\right. \\ \left. \times \cos 2\pi\tilde{\theta}_+ - g_{0,2}\cos 2\pi\tilde{\theta}_- - g_{1,1}\cos\pi\tilde{\theta}_+\cos\pi\tilde{\theta}_-\right) \tag{14}$$

The dual bosonic fields $\tilde{\theta}_+$ and $\tilde{\theta}_-$ describe the vortices of the fields $\theta_+$ and $\theta_-$. $g_{2,0}$, $g_{0,2}$, and $g_{1,1}$ are coupling parameters proportional to the fugacities of different types of vortices ($g_{2,0}/g_{0,2}$: integer vortices; $g_{1,1}$: half vortices).

The phase diagram obtained by the one-loop RG analysis provided in "Methods" is shown in Fig. 2a. Variation of the initial coupling parameters does not change the topology of the phase diagram, which

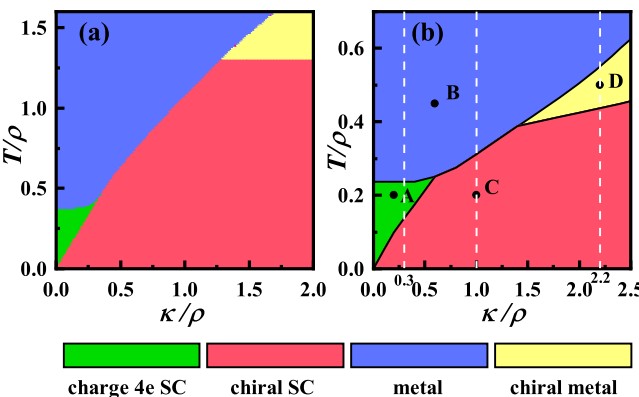

**Fig. 2 | The phase diagram.** Phase diagram provided by **a** the RG study and **b** the MC study. The initial values of the coupling parameters in **a** are $g_{2,0} = g_{0,2} = 0.1$, $g_{1,1} = g_4 = 0.01$ in Eq. (14) and in (**b**) are $A = 0.025\rho$ and $\gamma = \frac{1}{4}\rho\kappa/(\rho+\kappa)$ in Eq. (15). The three dotted lines and four dots (A–D) in (**b**) are used for subsequent interpretation of the phase diagram.

always includes the chiral TSC, charge-4e SC, chiral metal, and normal metal phases, see the Section IV of SI. At low enough $T$, the vortex fugacities $g_{2,0}$, $g_{0,2}$, and $g_{1,1}$ are all irrelevant while the IJC parameter $g_4$ is relevant, suggesting that both the $\theta_\pm$ fields are locked, leading to the TRS breaking chiral SC. With the enhancement of $T$, in the low $\kappa/\rho$ regime, $g_{0,2}$ first gets relevant (and suppresses $g_4$) suggesting that the $\theta_-$ vortices proliferate to restore the TRS, to form the charge-4e SC. In the high $\kappa/\rho$ regime instead, the $g_{2,0}$ first gets relevant suggesting the $\theta_+$ vortices proliferate to kill the SC, to form the chiral metal. In both regimes, at high enough $T$, $g_{2,0}$, and $g_{0,2}$ are both relevant, forming the normal metal phase. In the regime $\kappa \approx \rho$, with the enhancement of $T$, the system transits into a phase wherein the coupling $g_{1,1}$ is relevant and the half vortices involving both fields proliferate to kill both (quasi) orders, suggesting that the system directly transit to the normal state.

In the charge-4e SC, the Josephson-coupling phase, i.e., $\theta_-$, is disordered. However, this phase should not be understood as a layer-decoupled charge-2e SC from each layer, as in this phase the pairing phase of each layer is also disordered. To remind you, the charge-4e SC proposed here only lives in the intermediate temperature above the $T_c$ of the pairing state, wherein each layer is no longer superconducting. In the chiral metal phase, the time-reversal symmetry breaking can be verified by the polar Kerr effect. Furthermore, there can be spontaneously generated inner magnetic field in the material, which can be detected by the muon spin resonance experiment.

## MC study

To perform the MC study, we discretize the Hamiltonian (12) on the square lattice to obtain

$$H = -\alpha\sum_{\langle ij\rangle}\cos[\theta_t(\mathbf{r}_i)+\theta_b(\mathbf{r}_i)-\theta_t(\mathbf{r}_j)-\theta_b(\mathbf{r}_j)] \\ -\lambda\sum_{\langle ij\rangle}\cos[\theta_t(\mathbf{r}_i)-\theta_b(\mathbf{r}_i)-\theta_t(\mathbf{r}_j)+\theta_b(\mathbf{r}_j)] \\ -\gamma\sum_{\langle ij\rangle}\cos[\theta_t(\mathbf{r}_i)-\theta_t(\mathbf{r}_j)] + \cos[\theta_b(\mathbf{r}_i)-\theta_b(\mathbf{r}_j)] \\ +A\sum_i\cos[2\theta_t(\mathbf{r}_i)-2\theta_b(\mathbf{r}_i)]. \tag{15}$$

Here $\langle ij\rangle$ represents nearest-neighbor bonding, and the positive coefficients $\alpha$, $\lambda$, and $\gamma$ satisfy,

$$\alpha = \frac{\rho - 2\gamma}{4},\ \lambda = \frac{\kappa - 2\gamma}{4}. \tag{16}$$

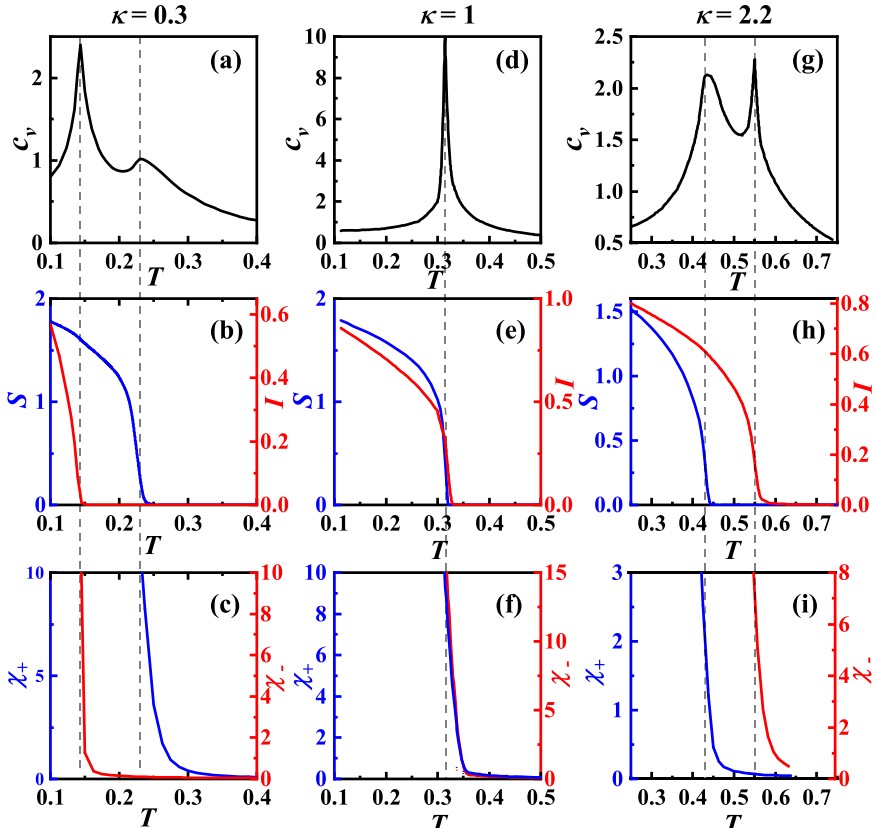

**Fig. 3 | Various T-dependent quantities.** Various T-dependent quantities for $\kappa = 0.3$ (**a–c**), $\kappa = 1$ (**d–f**), and $\kappa = 2.2$ (**g–i**). **a, d, g** The specific heat $C_v$. **b, e, h** The phase stiffness $S$ (blue) and Ising ODP $I$ (red). **c, f, i** The susceptibilities $\chi_+$ (blue) and $\chi_-$ (red). The $\rho$ is set as the unit of $\kappa$ and $T$.

Note that although different $\alpha$, $\lambda$, and $\gamma$ satisfying Eq. (16) leads to the same continuous Hamiltonian (12) in the continuum limit, it is required that all of them should be positive so as to reproduce the correct low-energy "classical Hilbert space" for allowed vortices. The reason is as follows. Here the $\alpha > 0$ and $\lambda > 0$ terms energetically allow for integer or half-integer $\theta_+$ and $\theta_-$ vortices, while the $\gamma > 0$ term energetically only allows for integer $\theta_t$ or $\theta_b$ vortices and hence imposes the "kinematic constraint" between the $\theta_+$ and $\theta_-$ vortices. Note that although the $\gamma$ term does not naturally emerge from Eq. (12), the singlevaluedness of the $\psi_{t/b}$ field dictates it. This term is crucial to yield the correct topology of the phase diagram. As shown in the Sec. VI of SI, if we turn off the $\gamma$ term, $\theta_+$ and $\theta_-$ are decoupled, leading to a topologically wrong phase diagram. A comparison between the correct phase diagram and the wrong one shows that the kinematic correlation makes the vestigial phase regimes largely shrink. For thermodynamic limit, even an infinitesimal $\gamma$ can energetically guarantee the "kinematic constraint". Here in the discrete lattice, we set $\gamma = \frac{1}{4}\rho\kappa/(\rho+\kappa)$, $A = 0.025\rho$, and their other values lead to similar results, see Section VI of SI.

The MC phase diagram shown in Fig. 2b is qualitatively consistent with the RG one shown in Fig. 2a. Various $T$-dependent quantities are shown in Fig. 3 for $\kappa/\rho = 0.3, 1, 2.2$ marked in Fig. 2b, with the formulas adopted in the MC calculations provided in "Methods". For $\kappa/\rho = 0.3$, the specific heat $C_v$ is shown in Fig. 3a, where the high-$T$ broad hump characterizes the Kosterlitz–Thouless (K–T) phase transition between the normal state and the charge-4e SC and the low-$T$ sharp peak characterizes the Ising phase transition between the charge-4e SC and the chiral SC. For this $\kappa/\rho$, Fig. 3b shows the phase stiffness $S$ characterizing the SC and the Ising ODP $I$ characterizing the relative-phase order[16], which emerge at the critical temperatures corresponding to the broad hump and sharp peak in

Fig. 3a, respectively. Furthermore, the total- ($\chi_+$) and relative- ($\chi_-$) phase susceptibilities[16] shown in Fig. 3c diverge at the same critical temperatures. For $\kappa/\rho = 1$, the specific heat shown in Fig. 3d exhibits only one peak, suggesting a direct phase transition from the normal state to the chiral SC. Such a result is also reflected in Fig. 3e, f which shows that the total- and relative-phase (quasi) orders emerge at the same temperature. For $\kappa/\rho = 2.2$, the corresponding results shown in Fig. 3g–i reveal that following the decrease of $T$, the system will successively experience the normal state, the chiral metal, and the chiral TSC phases. The results presented in Fig. 3 are well consistent with the phase diagram shown in Fig. 2b.

The total- (+) and relative- (−) phase correlation functions $\eta_\pm$ are shown in Fig. 4. See their formulas in "Methods". Figure 4a, b shows that for the representative point A marked in Fig. 2b, while $\eta_+(\Delta\mathbf{r})$ power-law decays with $\Delta r$ suggesting quasi-long-range order of the total phase, $\eta_-(\Delta\mathbf{r})$ decays exponentially with $\Delta r$, suggesting disorder of the relative phase. Obviously, these electron correlations are consistent with the charge-4e SC phase. Figure 4c, d shows that for the point D, while $\eta_+(\Delta\mathbf{r})$ decays exponentially with $\Delta r$ suggesting disorder of the total phase, $\eta_-(\Delta\mathbf{r})$ saturates to a constant number for large enough $\Delta r$ suggesting long-range order of the relative phase, consistent with the chiral-metal phase. For comparison, the $\eta_\pm$ for points B and C provided in Section V of SI is also consistent with the normal-metal and chiral-SC phases.

## Discussions

In comparison with previous proposals for the charge-4e/6e SC based on melting of the PDW[10,11,14] or the nematic pairing[17,18], our proposal is based on a more definite and easily realized start point: here we only need to start from non-topological $d$-wave SC (or $f$-wave SC) in any fourfold (or sixfold) symmetric monolayers. Particularly,

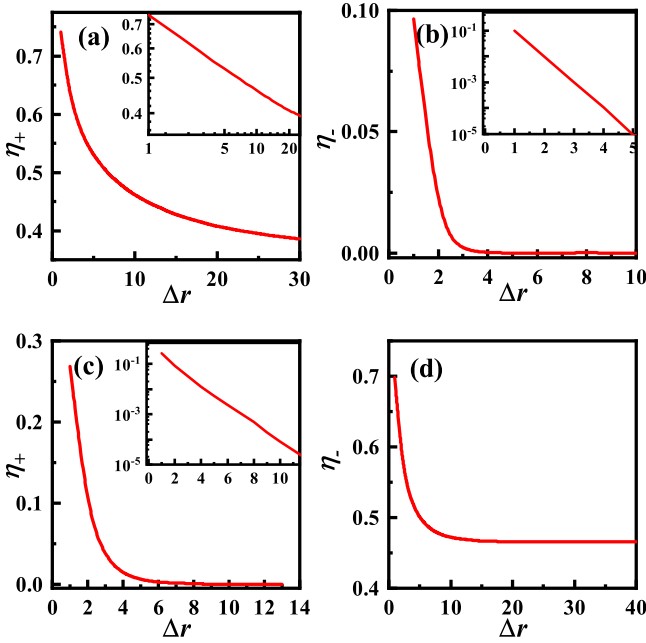

**Fig. 4 | The correlation functions.** The correlation function $\eta_\pm$ for **a** and **b** for A($\kappa = 0.2$, $T = 0.2$), and for **c** and **d** for D($\kappa = 2.2$, $T = 0.5$) marked in Fig. 2b. Insets of **a** the log–log plot, and **b** and **c** only the $y$-axes are logarithmic.

we have provided concrete synthesized materials to realize our proposal, i.e., the $45^o$-twisted bilayer cuprates and the $30^o$-twisted bilayer of some graphene family. Furthermore, superior to the previous bilayer approach, here a Cooper pair from the top layer pairs with a Cooper pair from the bottom layer to form the charge-4e SC between the layers. Consequently, the half flux quantization can be experimentally detected as a hallmark of the charge-4e SC in our proposal.

The TB-QC provides a better platform to realize the vestigial phases than conventional chiral superconductors such as the $p + ip$ or $d + id$ ones on the square or honeycomb lattices. The latter also hosts two degenerate pairing ODPs and hence can accommodate both total and relative-phase fluctuations of the two ODPs. However, the rotational symmetry of the monolayer system is not as high as that of the TB-QC studied here. Consequently, for chiral TSC in monolayer systems, there can be many nonzero coefficients in Eq. (9). Particularly, the two-phase fields are generally dynamically coupled as the symmetries in these systems allow for extra terms such as $\nabla_\pm \theta_+ \cdot \nabla_\pm \theta_-$ in the Hamiltonian density in Eq. (12). See more details in Section II of SI. As shown in Fig. 2 and Fig. S5a, the kinematic correlation between $\theta_+$ and $\theta_-$ has already made the vestigial phase regimes largely shrink, their extra dynamic coupling might make them further shrink or even vanish.

In conclusion, we have predicted the realization of the charge-4e SC or the chiral metal in the TB-QC, emerging as the unilateral (quasi) ordering of the total- or relative- pairing phase of the two layers, above the chiral-TSC ground state. The TB-QC provides a

better platform to realize these vestigial phases than previous proposals as here we can start from a more definite and easily realized start point.

## Methods
### The RG approach
Here we provide some technique details for the RG study. With standard RG analysis, the flow equations at the one-loop level are given by:

$$
\begin{aligned}
\frac{dg_{2,0}}{d\ln b} &= (2 - \pi\rho')g_{2,0} \\
\frac{dg_{0,2}}{d\ln b} &= (2 - \pi\kappa')g_{0,2} \\
\frac{dg_{1,1}}{d\ln b} &= \left(2 - \frac{\pi}{4}(\rho' + \kappa')\right)g_{1,1} \\
\frac{dg_4}{d\ln b} &= \left(2 - \frac{4}{\pi\kappa'}\right)g_4 \\
\frac{d\rho'}{d\ln b} &= -16g_{2,0}^2\rho'^3 - \frac{g_{1,1}^2}{2}\rho'^2(\rho' + \kappa') \\
\frac{d\kappa'}{d\ln b} &= \frac{256g_4^2}{\pi^4\kappa'} - 16g_{0,2}^2\kappa'^3 - \frac{g_{1,1}^2}{2}\kappa'^2(\rho' + \kappa'),
\end{aligned}
\tag{17}
$$

Here $b$ represents the renormalization scale, $g_{2,0}$, $g_{0,2}$, and $g_{1,1}$ represent the coupling strength of different types of topological defects, $\rho' = \rho/T$ and $\kappa' = \kappa/T$ represent two kinds of stiffness parameters.

The fixed points of N general RG flow equation $\frac{d\mathbf{g}}{d\ell} = R(\mathbf{g})$ is obtained by $R(\mathbf{g}^*) = 0$. In Table 1, we present four fixed points of the RG flow equation (17) and the corresponding phases in our calculation. We find the renormalized values of the stiffness parameters $\rho'$ and $\kappa'$ are consistent with the phase revealed by the RG flow result of the g-couplings. Specifically, the $\rho'$ flows to a finite positive value if the U(1)-gauge symmetry is (quasi) broken, otherwise it flows to zero; the $\kappa'$ flows to infinity if the time- reverse symmetry is broken, otherwise it flows to zero. Furthermore, the stability analysis of the fixed points can be provided following the standard process[83]. We only outline here and see more details in the Section III of SI. The $\beta$ function of the coupling constant which is very close to the fixed point $\mathbf{g}^*$ can be replaced by a linear mapping:

$$
R(\mathbf{g}) = R((\mathbf{g} - \mathbf{g}^*) + \mathbf{g}^*) \simeq M(\mathbf{g} - \mathbf{g}^*)
\tag{18}
$$

where we have used $R(\mathbf{g}^*) = 0$, and $M_{\alpha\beta} = \frac{\partial R_\alpha}{\partial g_\beta}|_{\mathbf{g}=\mathbf{g}^*}$. To get the stability properties of the flow, we have to diagonalize the matrix $M_{N\times N}$. The eigenvalues denoted by $\lambda_\alpha$, $\alpha = 1, 2, \ldots, N$. If the real parts of all the eigenvalues are negative or, at worst, zero, i.e., the scaling fields are all irrelevant or marginal. There are stable fixed points corresponding to the "stable phases". Complementary to the stable fixed points, if all the eigenvalues are positive and the scaling fields are all relevant, there are unstable fixed points. Additionally, there is a generic class of fixed points with both relevant and irrelevant scaling fields. These points are associated with the boundary of the phase transition.

### The Monte-Carlo approach
Here we provide some formulas for the MC calculations.

The phase stiffness characterizes the quasi-long-range order of the total phase and hence the SC is[16]

$$
S = \frac{1}{N}\left(\langle H_x \rangle - \beta \langle I_x^2 \rangle\right)
\tag{19}
$$

## Table 1 | Fixed points of the coupling parameters under RG, and the corresponding phases

| $g_{2,0}$ | $g_{0,2}$ | $g_4$ | $g_{1,1}$ | $\rho'$ | $\kappa'$ | Phase |
|---|---|---|---|---|---|---|
| $\infty$ | $\infty$ | 0 | $\infty$ | 0 | 0 | Normal |
| 0 | $\infty$ | 0 | 0 | $>8/\pi$ | 0 | Charge 4e SC |
| 0 | 0 | $\infty$ | 0 | $>2/\pi$ | $\infty$ | Chiral SC |
| $\infty$ | 0 | $\infty$ | 0 | 0 | $\infty$ | Chiral metal |

with

$$H_x = 4\alpha \sum_{<ij>_x} \cos[\theta_t(\mathbf{r}_i) + \theta_b(\mathbf{r}_i) - \theta_t(\mathbf{r}_j) + \theta_b(\mathbf{r}_j)]$$
$$+ \gamma \sum_{<ij>_x} \cos[\theta_t(\mathbf{r}_i) - \theta_t(\mathbf{r}_j)] + \cos[\theta_b(\mathbf{r}_i) - \theta_b(\mathbf{r}_j)]$$
$$I_x = 2\alpha \sum_{<ij>_x} \sin[\theta_t(\mathbf{r}_i) + \theta_b(\mathbf{r}_i) - \theta_t(\mathbf{r}_j) + \theta_b(\mathbf{r}_j)]$$
$$+ \gamma \sum_{<ij>_x} \sin[\theta_t(\mathbf{r}_i) - \theta_t(\mathbf{r}_j)] + \sin[\theta_b(\mathbf{r}_i) - \theta_b(\mathbf{r}_j)], \tag{20}$$

where $N$ is the site number, and $\beta = 1/k_B T$.

The Ising order parameter characterizing the relative-phase ordering breaking the time-reversal symmetry is,

$$I \equiv \frac{1}{N^2} \sum_{ij} \left\langle \sin[\theta_t(\mathbf{r}_i) - \theta_b(\mathbf{r}_i)] \cdot \sin[\theta_t(\mathbf{r}_j) - \theta_b(\mathbf{r}_j)] \right\rangle.$$

The total- (+) and relative- (−) phase susceptibilities for temperatures above the $T_c$ of the corresponding orders are defined by

$$\chi_{\pm} \equiv \frac{1}{NT} \sum_i \left\langle \left| e^{i[\theta_t(\mathbf{r}_i) \pm \theta_b(\mathbf{r}_i)]} \right|^2 \right\rangle. \tag{21}$$

The total- (+) and relative- (−) phase correlation functions are defined as

$$\eta_{\pm}(\Delta\mathbf{r}) = \frac{1}{N} \sum_{\mathbf{r}} \left\langle e^{i[\theta_t(\mathbf{r}) \pm \theta_b(\mathbf{r}) - \theta_t(\mathbf{r}+\Delta\mathbf{r}) \mp \theta_b(\mathbf{r}+\Delta\mathbf{r})]} \right\rangle. \tag{22}$$

## Data availability
All data are displayed in the main text and Supplementary Information.

## Code availability
The code that supports the plots within this paper is available from the corresponding author upon request.

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

## Acknowledgements

We are grateful to the stimulating discussions with Zhi-Ming Pan, Shao-Kai Jian, Chen Lu, Meng Zeng, and Wei-Qiang Chen. F.Y. is supported by the National Natural Science Foundation of China under the Grant Nos. 12074031 and 11674025. J.Z. is supported by the funding of the Institute for Advanced Sciences of Chongqing University of Posts and Tele-communications (E011A2022326). C.W. is supported by the National Natural Science Foundation of China under the Grant Nos. 12234016 and 12174317. This work has been supported by the New Cornerstone Science Foundation.

## Author contributions

F.Y. proposed the main idea and supervised the study. Y.-B.L. performed the MC study. J.Z. performed the RG study. All authors significantly contributed to the data analysis and paper writing.

## Competing interests

The authors declare no competing interests.
