## [Peer Review File · Nature Communications]

Charge-4e superconductivity and chiral metal in 45°-twisted bilayer cuprates and similar bilayersReviewers' Comments:

Reviewer #1:

Remarks to the Author:

The authors have conducted an investigation into possible charge- $4e$ states in twisted bilayer materials. At low temperatures, each monolayer exhibits superconductivity (SC) with the largest pairing angular momentum, such as the cuprate monolayer with C_4 symmetry hosting d-wave SC. When two monolayers with D_n symmetry are stacked at the largest twist angle π/n , an additional symmetry emerges---a π/n rotation followed by an exchange of the top and bottom layers.

This symmetry crucially prohibits the first-order interlayer Josephson coupling (IJC). Under certain conditions, the second-order IJC favors a chiral SC order that breaks time-reversal symmetry through an unusual ϕ locking between the order parameters in the two layers. As the ground state breaks both time-reversal and $U(1)$ symmetry, it becomes promising to realize the charge- $4e$ SC when only the time-reversal symmetry is restored.

Starting from the chiral SC state, the authors explore possible vestigial orders as the temperature is increased. They carefully derive the Landau-Ginzburg free energy and employ two comprehensive methods, renormalization group and Monte Carlo simulation, to analyze its different phases. Through a detailed analysis of topological defects in various regimes, they convincingly demonstrate that the charge- $4e$ SC occupies a substantial region in the phase diagram. While similar calculations have been performed in previous papers, the relevance of this study to materials realization adds a unique and important aspect to their proposal.

However, a comment regarding Eq. (13) arises, as the term $\cos 4\theta_-$ seems to imply the existence of four degenerate ground states at $\theta_- = \pm \pi/4$ and $\pm 3\pi/4$. This contradicts the Ising nature of $d \pm id$ states and the subsequent numerical data. It is essential for the authors to address this point and provide an explanation.

Furthermore, the authors support their findings with a Monte Carlo simulation by discretizing the Hamiltonian on a square lattice. The numerical data strongly suggests the presence of a charge- $4e$ SC phase in an intermediate temperature range. This strengthens the overall conclusion of the study. Given the novel proposal, comprehensive analysis of the intriguing charge- $4e$ SC order, and the potential resolution of the comment mentioned above, I believe this paper will generate significant interest among a broad range of readers in the scientific community. Therefore, I recommend its publication in Nature Communications, provided the authors address the comment raised.

Reviewer #2:

Remarks to the Author:

The high- T_c cuprate has a single-component charge- $2e$ pairing order parameter of d-wave symmetry, which is time reversal invariant. Recently, there are both theoretical and experimental interests in the so-called 45-degree twisted bilayer cuprates, where the Josephson coupling between the two 45-degree rotated d-wave order parameters has been proposed to generate a time-reversal symmetry breaking superconductor with $d+id$ symmetry. Experimental realization of such a chiral superconductor has been challenging because of the difficulty in reaching the ideal Josephson coupling due to, for example, disorder and inhomogeneity of the cuprate superconductors.

In this paper, assuming the twisted bilayer chiral superconductor is realized, the authors study the thermal melting of the composite state. They propose that charge- $4e$ superconductivity (as well as a chiral metal phase) can arise as an intermediate vestigial ordered phase. Charge $4e$ and charge $6e$ paired states have long been proposed and sought after. The recent experimental evidence for charge $4e$ and $6e$ quantization in certain kagome superconductors have triggered renewed interests and brought them closer to light. Thus, the subject matter of this manuscript is interesting and timely and the present paper can in principle contribute to these rapidly developing fields. However, there are a few scientific issues at the current stage that must be addressed and

clarified, before I can reach a decision with regard to whether the paper meets the criteria for publication in Nature Communications.

1. The authors compare their work to other proposals of vestigial charge $4e$ order in the introduction paragraphs. Some of these discussions are not well constructed and also inaccurate. First, in theory, the difference between incommensurate and commensurate PDWs is about continuous versus discrete symmetries, which are not crucial when considering the melting of the translation symmetry breaking for vestigial order. The incommensurate PDW is usually pinned by defects and disorder, and a commensurate PDW can have very small locking energies, especially for higher commensurations depending on the details. Second, the PDW order has already been observed in several systems. One should make the discussions clearer, since the true problem with the PDW scenario for charge $4e$ superconductivity is that so far the observed PDW order is not a pure PDW order; there is always a dominant uniform SC part. Thus, it is more advantageous to work with a dominant SC order that is nevertheless composite for vestigial $4e$ state. In this sense, nematic superconductors and chiral superconductors studied here, are more realistic within these considerations. Third, it is not accurate to say that nematic superconductors are harder to realize than chiral superconductors in the 2 dimensional representation, since this depends only on model or system parameters. Considering the available experiments, there are actually more evidence for nematic superconductors than those for chiral superconductors. Theoretically, the underlying mechanism and phase structure of melting in these two scenarios are nearly identical, as can be seen from the action in Eq. (13) and the Sine-Gordon model for the dual fields in Eq. (14). Finally, as far as I understand, the melting of the multi-component hexatic chiral superconductor leading to vestigial charge $6e$ superconductivity was proposed in Ref. 22 in the context of kagome superconductors.

I recommend that the authors revise their introductory and concluding paragraphs with more fair and appropriate discussions. The current theoretical and experimental situation is not at the stage to discriminate one scenario over the other. Doing so does not diminish the importance of the current work, only to provide the readers with a clear and accurate exposition of the development in the field.

2. Concerning the theory part, as I alluded above, the phase structure is known and the RG study of the dual action in Eq. (14) is nearly identical to previous published works. A more recent work is arXiv:2303.00653, where different kinds of vestigial order from nematic and chiral superconductors are discussed on equal footing. The strength of the current manuscript is in the Monte Carlo simulations of the phase-only model discretized on the square lattice in Eq. (15). The results are indeed very impressive. Naturally, it is important to explain how the lattice model truly represents the phase-only continuum model in Eq. (13), since different lattice models can all reduce to Eq. (13) in the continuum limit. In this regard, the gamma-term introduced in Eq. (15) needs to be discussed in more detail and better justified. This term does not naturally come out of the theory. In the manuscript, the author argues that this term is introduced to enforce the singlevaluedness of the order parameters, but this is counterintuitive since they introduce compact polynomials of the gradients in θ_+ and θ_- , whereas the kinematics of the phases should arise from the theory as a requirement of internal self-consistency. Therefore, the introduction of the gamma-term and its effects must be discussed more carefully in the manuscript and studied numerically. For example, the effective lattice model of the twisted bilayer d-wave superconductors can be written down straightforwardly. Can the gamma-term be seen and derived from such a microscopic model? Without this, the gamma term can only be considered as a numerical stabilization path, and the limit gamma goes to 0 should be studied and discussed. Also, the effects of the four-fold anisotropy, and how the results depend on the choice of A in the Monte Carlo results should be included.

3. The authors argue that the intermediate phase with exponential decay of the correlation in the relative phase θ_- , but power law decay in the total phase θ_+ , is the charge- $4e$ phase. In this region, the phase of the Josephson coupling in Eq. (12) is also disordered, making the interlayer coupling short-range correlated spatially. The authors should discuss the physical consequences of this, e.g. how this is related or unrelated to the "decoupled" charge- $2e$ d-wave state from each layer.

Reviewer #3:

Remarks to the Author:

The authors of 'Charge-4e superconductivity and chiral metal in the 45°-twisted bilayer cuprates and similar materials' have thoroughly examined the mechanisms underlying the elusive charge-4e superconductivity and have proposed experimental realisations. The work is well-argued and communicated. The introduction successfully establishes the significance of the topic, and the phenomenological modelling and employed techniques are both intriguing and well-presented. I am particularly persuaded by the combination of the symmetry analysis, renormalization group, and numerical Monte Carlo, which together provide confidence in the findings. The suggested physical systems are sensible, topical and attainable experimentally. This publication undoubtedly constitutes a valuable contribution to the existing literature.

Before providing a definitive recommendation, I would appreciate it if the authors could address the following comments:

Comment 1

(i) How/where do you account the constraint on θ_+ and θ_- ?

(ii) There is a statement in the paper: ^[1]_{SEP}

"As shown in Fig. 2, the kinematic correlation between θ_+ and θ_- has already made the vestigial phase regimes largely shrink, their extra dynamic coupling might make them further shrink or even vanish" ^[1]_{SEP}

I think the authors should elaborate on this statement. Namely, did you compute this phase diagram both with and without the constraint? (Is that the content of Fig. S3?) Or is there some other way to understand your statement? ^[1]_{SEP}

(iii) Relatedly, a more clear description of the statement — i.e that coupling between θ_+ and θ_- is disfavours the desired phases — is rather important for the logic of the paper since it is used to contrast this proposal from previous proposals based on chiral TSC in monolayers systems. On this point, I think the paper would benefit from some additional contrast between the proposed twisted models and the previously studied un-twisted models for chiral condensates.

Comment 2

Is the γ parameter in Fig. 2 the same γ used in Eq. (9) and/or (15)? I ask because the γ in Eq. (9) should be zero by symmetry, right? Also, for Fig. 2 I think it would be helpful to state that the RG is performed using Eq. (14) and the monte Carlo using Eq. (15). If that is indeed correct. ^[1]_{SEP}

Comment 3

I think that a sentence or two describing the Chiral metal would be very helpful for the reader. Although the authors do provide references, I think it would make for an easier read to just define what this phase is. ^[1]_{SEP}

Comment 4

(i) Why does the fixed point table only include the g-couplings? Is it assumed that all other couplings (ρ and κ) are zero or infinity or something else?

^[1]_{SEP}(ii) On the fixed points -- which of these are stable/unstable? It would be interesting to add this detail. For example, Eq. (5) and onwards of the work [Park, Ye and Balents Phys. Rev. B 104, 035142 (2021)], outlines a pretty clear method to compute stability of fixed points.

Comment 5

There are a few typos throughout the main tex that should be addressed. Also, in the Supplement, I suspect Eq. (9) has a typo: should it contain a v_{ij} factor instead of one of the γ_{ij} ?

Summary

The paper is exciting, thorough and well argued. To improve, I think it would be beneficial to provide a clearer differentiation between the realisation of charge-4e superconductivity in the twisted bilayer quasicrystal scheme and the existing proposals in monolayers. Perhaps addressing 'Comment 1' mentioned above would already suffice in achieving this clarity.

Best regards,
Harley Scammell

REVIEWER1

Comment: The authors have conducted an investigation into possible charge-4e states in twisted bilayer materials. At low temperatures, each monolayer exhibits superconductivity (SC) with the largest pairing angular momentum, such as the cuprate monolayer with C_4 symmetry hosting d-wave SC. When two monolayers with D_n symmetry are stacked at the largest twist angle π/n , an additional symmetry emerges— a π/n rotation followed by an exchange of the top and bottom layers. This symmetry crucially prohibits the first-order interlayer Josephson coupling (IJC). Under certain conditions, the second-order IJC favors a chiral SC order that breaks time-reversal symmetry through an unusual $\pm i$ locking between the order parameters in the two layers. As the ground state breaks both time-reversal and $U(1)$ symmetry, it becomes promising to realize the charge-4e SC when only the time-reversal symmetry is restored.

Response: We thank the Reviewer very much for the comments! These comments accurately summarize the motivation and starting point of our work, reflecting her/his professionalism in this research field.

Comment: Starting from the chiral SC state, the authors explore possible vestigial orders as the temperature is increased. They carefully derive the Landau-Ginzburg free energy and employ two comprehensive methods, renormalization group and Monte Carlo simulation, to analyze its

different phases. Through a detailed analysis of topological defects in various regimes, they convincingly demonstrate that the charge-4e SC occupies a substantial region in the phase diagram. While similar calculations have been performed in previous papers, the relevance of this study to materials realization adds a unique and important aspect to their proposal.

Response : Thanks a lot to the Reviewer for the comments! These comments nicely summarize the main contribution of our work. We particularly appreciate the Reviewer for the comment '*the relevance of this study to materials realization adds a unique and important aspect to their proposal.*'

Comment: However, a comment regarding Eq. (13) arises, as the term $\cos(4\theta_-)$ seems to imply the existence of four degenerate ground states at $\theta_- = \pm\pi/4$ and $\pm 3\pi/4$. This contradicts the Ising nature of $d \pm id$ states and the subsequent numerical data. It is essential for the authors to address this point and provide an explanation.

Response: We thank the Reviewer very much for the comment, which provides us with the opportunity to clear the misunderstanding and to enhance the presentation of our manuscript.

As pointed out by the Reviewer, the term $\cos(4\theta_-)$ in Eq. (13) does lead to

four different values of θ_- : $\theta_- = \pm\pi/4$ and $\pm 3\pi/4$ for the ground state. However, only two out of the four are gauge inequivalent. To elucidate this point, let us take the ground states represented by $\theta_- = \pi/4$ and $\theta_- = -3\pi/4$ as an example. From Eq. (7), we have $\theta_t = \theta_+ + \theta_-$, $\theta_b = \theta_+ - \theta_-$, which yields $\theta_t + \theta_b = 2\theta_+$, $\theta_t - \theta_b = 2\theta_-$. This relation suggests that θ_- represents half of the phase difference between the two layers. Therefore, the $\theta_- = \pi/4$ and $\theta_- = -3\pi/4$ actually represent the same physical angle $\theta_t - \theta_b = \pi/2, -3\pi/2$, because two angles different by 2π are physically equal. It seems that for the two different values of θ_- , i.e. $\theta_- = \pi/4$ and $\theta_- = -3\pi/4$, the corresponding values of θ_t differ by $-\pi$ and those of θ_b differ by π . However, since $-\pi$ and π are physically equal, this difference only represents a global phase change. Physically, pairing states different solely by a global phase change are gauge equivalent, i.e. can be related by a global U(1)-gauge rotation. Here, when we speak of the ground state degeneracy, we actually mean the degeneracy in addition to the U(1) degeneracy brought about by the global U(1)-gauge symmetry breaking. Thus, the gauge equivalent states should not be counted in. Therefore, the $\theta_- = \pi/4$ and $\theta_- = -3\pi/4$ actually label gauge equivalent states and cannot be counted as two degenerate ground states, and so do $\theta_- = -\pi/4$ and $\theta_- = 3\pi/4$. However, the $\theta_- = \pi/4$ and $\theta_- = -\pi/4$ indicate two gauge inequivalent states with physical angle $\theta_t - \theta_b = \pi/2$, and $-\pi/2$, which are time reversal related, and so do $\theta_- = -3\pi/4$ and $\theta_- = 3\pi/4$. Therefore, we have only two

gauge inequivalent degenerate ground states, which are time-reversal partners. This explains the Ising nature of the d+id SC.

In the numerical aspect, let us take the case of $\kappa/\rho=0.3$ displayed in Fig. 2(b) and Fig. 3(a-c) as an example to clarify the point. At sufficiently low temperatures, Eq. (13) yields that θ_+ and θ_- are both pinned down. The θ_+ is pinned down randomly at an arbitrary value, and change of this value will only lead to gauge equivalent states. The θ_- is pinned down randomly at one out of the four values: $\theta_- = \pm\pi/4$ and $\pm 3\pi/4$. As clarified on the above, only two out of the four are gauge inequivalent. Considering the combined (θ_+, θ_-) configuration, the system hosts two gauge inequivalent degenerate ground states related by time-reversal symmetry, which means that the ground state symmetry breaking is $U(1)*Z_2$. This represents the d+id SC displayed by the pink-colored regime in Fig. 2(b). When the temperature arises above the T_c of the d+id SC, the θ_+ is still pinned down randomly at an arbitrary value, while the θ_- dynamically fluctuate among its different saddle points. In such a case, the system hosts only one gauge inequivalent state, meaning that the symmetry breaking is $U(1)$. This represents the charge-4e SC phase displayed by the green-colored regime in Fig. 2(b). Therefore, the phase transition from the d+id SC with $U(1)*Z_2$ symmetry breaking to the charge-4e SC with $U(1)$ symmetry breaking is a Z_2 Ising-type phase transition characterized by time-reversal

symmetry breaking. This Ising transition is featured by the specific heat peak displayed in Fig. 3(a), the vanish of the Ising order parameter displayed in Fig. 3(b) and the divergence of the Ising susceptibility displayed in Fig. 3(c). This explains the numerical data.

This misunderstanding proposed by the Reviewer can also be cleared if we redefine θ_+ and θ_- as $\theta_+ = \theta_t + \theta_b$, $\theta_- = \theta_t - \theta_b$. Under this redefinition, the $\cos(4\theta_-)$ term in Eq.(13) will be changed to $\cos(2\theta_-)$. The Ising nature of θ_- is now more intuitive, but the subsequent conclusions do not change.

According to this comment, in the revised manuscript, we supplement an explanation of the Ising nature of θ_- .

Comment: Furthermore, the authors support their findings with a Monte Carlo simulation by discretizing the Hamiltonian on a square lattice. The numerical data strongly suggests the presence of a charge-4e SC phase in an intermediate temperature range. This strengthens the overall conclusion of the study. Given the novel proposal, comprehensive analysis of the intriguing charge-4e SC order, and the potential resolution of the comment mentioned above, I believe this paper will generate significant interest among a broad range of readers in the scientific community. Therefore, I recommend its publication in Nature

Communications, provided the authors address the comment raised.

Response: Thanks a lot to the Reviewer! We appreciate the Reviewer very much for her/his high assessment of our work.

We also thank the Reviewer for proposing the comment, which provides us with the opportunity to clear the draw back in the presentation of our manuscript. According to this comment, we have described the Ising nature of θ in more details to avoid possible misunderstandings. We believe that the revised manuscript now meets the high standard of Nature Communications.

REVIEWER2

Comment: In this paper, assuming the twisted bilayer chiral superconductor is realized, the authors study the thermal melting of the composite state. They propose that charge-4e superconductivity (as well as a chiral metal phase) can arise as an intermediate vestigial ordered phase. charge 4e and charge 6e paired states have long been proposed and sought after. The recent experimental evidence for charge 4e and 6e quantization in certain kagome superconductors have triggered renewed interests and brought them closer to light. Thus, the subject matter of this manuscript is interesting and timely and the present paper can in principle contribute to these rapidly developing fields. However, there are a few

scientific issues at the current stage that must be addressed and clarified, before I can reach a decision with regard to whether the paper meets the criteria for publication in Nature Communications.

Response: We thank the Reviewer for the accurate summarize of our work! We appreciate her/him very much for the comment “*Thus, the subject matter of this manuscript is interesting and timely and the present paper can in principle contribute to these rapidly developing fields.*” We are also grateful to the Reviewer for proposing the several scientific issues, which provides us with the opportunity to clarify them and hence to enhance the quality of our manuscript.

Comment 1:

Comment: The authors compare their work to other proposals of vestigial charge 4e order in the introduction paragraphs. Some of these discussions are not well constructed and also inaccurate..... I recommend that the authors revise their introductory and concluding paragraphs with more fair and appropriate discussions. The current theoretical and experimental situation is not at the stage to discriminate one scenario over the other. Doing so does not diminish the importance of the current work, only to provide the readers with a clear and accurate exposition of the development in the field.

Response: We thank the Reviewer very much for pointing out the many

drawbacks in the presentation of our manuscript, as well as proposing a lot of constructive revision suggestions for the manuscript! We well agree with all the viewpoints proposed by the Reviewer and have revised the manuscript according to her/his suggestions. Concretely, we have revised the introduction to fairly introduce the existing approaches to realize the charge-4e SC. Particularly, we have highlighted Ref. [22], as the proposed approach there to realize the charge-6e SC in the Kagome superconductors is related to our work. We have also revised the conclusion part to fairly summarize the contribution of our work.

Comment 2:

Concerning the theory part, as I alluded above, the phase structure is known and the RG study of the dual action in Eq. (14) is nearly identical to previous published works. A more recent work is arXiv:2303.00653, where different kinds of vestigial order from nematic and chiral superconductors are discussed on equal footing. The strength of the current manuscript is in the Monte Carlo simulations of the phase-only model discretized on the square lattice in Eq. (15). The results are indeed very impressive. Naturally, it is important to explain how the lattice model truly represents the phase-only continuum model in Eq. (13), since different lattice models can all reduce to Eq. (13) in the continuum limit. In this regard, the gamma-term introduced in Eq. (15) needs to be

discussed in more detail and better justified. This term does not naturally come out of the theory. In the manuscript, the author argues that this term is introduced to enforce the singlevaluedness of the order parameters, but this is counterintuitive since they introduce compact polynomials of the gradients in θ_+ and θ_- , whereas the kinematics of the phases should arise from the theory as a requirement of internal self-consistency. Therefore, the introduction of the gamma-term and its effects must be discussed more carefully in the manuscript and studied numerically. For example, the effective lattice model of the twisted bilayer d-wave superconductors can be written down straightforwardly. Can the gamma-term be seen and derived from such a microscopic model? Without this, the gamma term can only be considered as a numerical stabilization path, and the limit gamma goes to 0 should be studied and discussed. Also, the effects of the four-fold anisotropy, and how the results depend on the choice of A in the Monte Carlo results should be included.

Response: Thanks a lot to the Reviewer for the comments! We are grateful to her/him for providing us with the useful reference, which has been added in the revised reference list. We appreciate very much her/his comment *“The strength of the current manuscript is in the Monte Carlo simulations of the phase-only model discretized on the square lattice in Eq. (15). The results are indeed very impressive.”* We agree with the Reviewer in that it is important to explain how the lattice model truly

represents the phase-only continuum model in Eq. (13). We particularly agree with her/him in that the γ term introduced in Eq. (15) is crucial, which is explained in detail in the following.

Suppose we turn off the γ term in Eq. (15), we have $\alpha=\rho/4$, $\lambda=\kappa/4$. Then we redo the Monte-Carlo calculations. Consequently, the obtained phase diagram is displayed in the following Fig. R1(a). This phase diagram is very simple, which is divided by two lines into four phases touching at one qua-critical point. The straight line parallel to the x-axis represents the K-T transition, suggesting that the (quasi-) ordering temperature of θ_+ only relies on ρ . The (nearly straight) line passing through the coordinate origin represents the Ising transition, suggesting that the ordering temperature of θ_- only relies on κ when fixing A. This phase diagram suggests that θ_+ and θ_- are decoupled, which is understood as follow.

Fig.R1. (a) Monte-Carlo phase diagram for $\gamma=0$. (b) Integral region of θ_b and θ_t at a given site. (c) Expanded integral region of θ_b and θ_t at a given site.

The partition function of the model is written as

$$Z = \int \dots \int \prod_{\vec{r}_i} d\theta_t(\vec{r}_i) d\theta_b(\vec{r}_i) e^{-\beta H[\{\theta_t(\vec{r}_i), \theta_b(\vec{r}_i)\}]} \quad (\text{R1})$$

For each site, the integral region is within the “first Brillouin Zone (BZ)”

$\theta_t \in [-\pi, \pi)$, $\theta_b \in [-\pi, \pi)$ shown in Fig. R1(b). As $H[\{\theta_t(\vec{r}_i), \theta_b(\vec{r}_i)\}]$

is a periodic function of θ_b and θ_t with period 2π , the integral region can

be expanded to the “second BZ” shown in Fig. R1(c), i.e.

$\theta_t + \theta_b \in [-2\pi, 2\pi)$, $\theta_t - \theta_b \in [-2\pi, 2\pi)$ or equivalently

$\theta_+ \in [-\pi, \pi)$, $\theta_- \in [-\pi, \pi)$. This expansion only doubles the

partition function, and would not change the physics. For $\gamma=0$, we have

$H = H_+[\{\theta_+(\vec{r})\}] + H_-[\{\theta_-(\vec{r})\}]$, and then

$$\begin{aligned} Z &= \int \dots \int \prod_{\vec{r}_i} d\theta_+(\vec{r}_i) d\theta_-(\vec{r}_i) e^{-\beta H_+[\{\theta_+(\vec{r}_i)\}]} \cdot e^{-\beta H_-[\{\theta_-(\vec{r}_i)\}]} \\ &= \int \dots \int \prod_{\vec{r}_i} d\theta_+(\vec{r}_i) e^{-\beta H_+[\{\theta_+(\vec{r}_i)\}]} \cdot \int \dots \int \prod_{\vec{r}_i} d\theta_-(\vec{r}_i) e^{-\beta H_-[\{\theta_-(\vec{r}_i)\}]} \\ &= Z_+ \cdot Z_- \end{aligned}$$

(R2)

This derivation explains why θ_+ and θ_+ are decoupled for $\gamma=0$.

However, the phase diagram shown in the above Fig. R1(a) is topologically different from Fig. 2 in the main text and is wrong. For comparison, we copy Fig. 2 and paste it here as Fig. R2. Physically, while in Fig. R1(a) the vestigial phases always emerge above the T_c of the d+id SC except for one isolate point, in Fig. R2 when $\kappa \approx \rho$ the system will transition from the d+id SC directly into the normal metal state. To clarify

this point, let us start from the continuous Hamiltonian Eq. (12). At a glance, this Hamiltonian also decouples the θ_+ and θ_- fields. However, the two fields are correlated with each other via a hidden “kinematic constraint”, which is explained as follow.

Fig. R2. The RG phase diagram.

Similar with previous studies in Ref. [10, 17, 18], the phase transitions here are determined by the proliferation of the vortices of the θ_+ and θ_- fields. Therefore, the possible types of vortices allowed to be excited constitute the low-energy “classical Hilbert space” of the continuous Hamiltonian. Labeling the winding numbers of the θ_+ and θ_- vortices as n_+ and n_- and those of the θ_t and θ_b vortices as n_t and n_b , we have

$$n_+ = \frac{n_t + n_b}{2}, \quad n_- = \frac{n_t - n_b}{2} \quad (\text{R3})$$

Here n_t and n_b for the physical angles θ_t and θ_b should be integers to guarantee the single-valuedness of the order parameters $\Delta_t(r)$ and $\Delta_b(r)$. Then n_+ and n_- should be integers or half integers simultaneously. This

sets as a “kinematic constraint” in the “classical Hilbert space” which correlates the θ_+ and θ_- fields. In other words, while the θ_+ and θ_- fields each can host either integer- or half-integer- vortices, the “kinematic constraint” requires that they should simultaneously host integer vortices or simultaneously host half-integer vortices. Under this “kinematic constraint”, there exist three types of vortices, i.e. the θ_+ vortices, the θ_- vortices and the half-half vortices. It is just the proliferation of the half-half vortices which drives the phase transition from the d+id SC directly into the normal state, resulting into the correct phase diagram Fig. R2. Therefore, the “kinematic constraint” between the θ_+ and θ_- fields is crucial in obtaining the correct topology of the phase diagram.

Then let us come back to the lattice model Eq. (15). Although Eq. (15) is reduced to the same formula as Eq. (12) in the continuum limit for any group of α , λ and γ satisfying Eq. (16), it is still needed that Eq. (15) can reproduce the correct “classical Hilbert space” for the allowed types of vortices in the continuum limit. Unlike in the continuous space wherein the “classical Hilbert space” originates from the single-valuedness of the physical fields Δ_t and Δ_b , in the discrete lattice the correct “classical Hilbert space” can only be self-consistently realized via free-energy minimization. For this purpose, let us investigate what types of vortices are energetically allowed by Eq. (15) in the continuum limit. Firstly, the γ

term with $\gamma > 0$ only energetically allows for integer θ_t or θ_b vortices. To clarify this, we schematically display an assumed non-integer vortex, say, a half vortex, of the θ_t or θ_b field in Fig. R3. In this configuration, there exists a singular branch cut (the red line): the phase angles on the two sides of the cut are different by π . For $\gamma > 0$, the energy cost for this configuration is proportional to γL , in which L indicates the lattice size. For large enough L , this free energy cost cannot be compensated by that gain through the entropy which scales with $\ln(L)$. Therefore, $\gamma > 0$ in Eq. (15) energetically requires integer n_t and n_b . Secondly, similarly, the α term with $\alpha > 0$ energetically allows for integer $n_t + n_b$, which requires that n_+ can be either integer or half integer. Thirdly, the λ term with $\lambda > 0$ energetically allows for integer or half-integer n_- . To summarize, while the combined α and λ terms energetically allow for integer or half-integer θ_+ or θ_- vortices, the γ term imposes the “kinematic constraint” between them via energy minimization. The α , λ , and γ terms combinedly reproduce the correct low-energy “classical Hilbert space” in the continuum limit. Then it is no wonder that the γ term is crucial to reproduce topologically correct phase diagram.

Fig. R3. A non-integer (half) vortex configuration of the θ_t or θ_b field.

Technically, it is not practicable to derive the lattice Hamiltonian Eq. (15) directly from the original electron Hamiltonian such as the t-J model. As introduced in, say “*Condensed Matter Field Theory (Second Edition)*” (A. Atland and B. Simons, 2010 Cambridge), the analytical form of the effective Hamiltonian for the order-parameter field can only be derived in the continuum limit, leading again to the continuous Hamiltonian Eq. (12). However, the single-valuedness of the pairing order parameter fields $\Delta_t(\mathbf{r})$ and $\Delta_b(\mathbf{r})$ always requires the emergence of the γ term or similar terms in the lattice Hamiltonian to guarantee that n_t and n_b are integers, because the α and λ terms cannot guarantee that. The above analysis based on Fig. R3 suggests that for large enough lattices, even an infinitesimal γ can energetically guarantee the “kinematic constraint”. However, for a finite

lattice, we need a weak but finite γ to guarantee that. In our calculations, we have tried different weak γ , and the obtained phase diagrams are topologically the same. In the limit of γ goes to zero, the phase diagram should be topologically the same in the thermal-dynamic limit.

In the revised manuscript, we have supplemented more detailed discussions about the γ term, both in the main text and the Supplementary Materials. We have also supplemented phase diagrams with different values of A in the Supplementary Materials, and we can see that the realistic range values of A do not change the topological structure of the phase diagram.

Comment 3:

The authors argue that the intermediate phase with exponential decay of the correlation in the relative phase θ_- , but power law decay in the total phase θ_+ , is the charge- $4e$ phase. In this region, the phase of the Josephson coupling in Eq. (12) is also disordered, making the interlayer coupling short-range correlated spatially. The authors should discuss the physical consequences of this, e.g. how this is related or unrelated to the "decoupled" charge- $2e$ d-wave state from each layer.

Response: Thanks to the Reviewer for the comments!

In the charge-4e SC phase, the correlation in the relative phase θ_- exponential decays, while that in the total phase θ_+ power law decays. In this phase, the phase of the Josephson coupling, i.e. θ_- , is disordered, making the interlayer coupling short-range correlated spatially. However, the total phase θ_+ is quasi ordered. This phase is possible because from our Ginzburg-Landau analysis, the two independent stiffness parameters ρ and κ can satisfy $\kappa < \rho$. In such cases, the time-reverse symmetry becomes easier to restore when increasing the temperature and the system enters the charge-4e SC phase.

The charge-4e SC phase is not directly related to the layer-decoupled charge-2e d-wave SC from each layer, because according to Eq. (7), when θ_- is disordered but θ_+ is ordered, the phase of each layer is also disordered. To remind, the charge-4e SC proposed here is a vestigial phase which lives in the intermediate temperature above the T_c of the pairing state. In this temperature region, each layer is no longer superconducting.

The above discussions have been supplemented to the revised manuscript.

REVIEWER3

Comment: The authors of 'Charge-4e superconductivity and chiral metal

in the 45°-twisted bilayer cuprates and similar materials' have thoroughly examined the mechanisms underlying the elusive charge-4e superconductivity and have proposed experimental realizations. The work is well-argued and communicated. The introduction successfully establishes the significance of the topic, and the phenomenological modelling and employed techniques are both intriguing and well-presented. I am particularly persuaded by the combination of the symmetry analysis, renormalization group, and numerical Monte Carlo, which together provide confidence in the findings. The suggested physical systems are sensible, topical and attainable experimentally. This publication undoubtedly constitutes a valuable contribution to the existing literature. Before providing a definitive recommendation, I would appreciate it if the authors could address the following comments:

Response: Thanks a lot to the Reviewer for the comments! We greatly appreciate the Reviewer for the high assessment of our work. We are also grateful to the comments proposed, which provides us with opportunity to enhance the presentation of our manuscript.

Comment 1:

Comment: (i) How/where do you account the constraint on θ_+ and θ_- ?

Response: Thanks a lot to the Reviewer for the comment!

The constraint on the relative phase θ_- and total phase θ_+ arises from the single-valuedness of the pairing order parameters Δ_t and Δ_b . As the phases of the physical quantities Δ_t and Δ_b , θ_t and θ_b can only change by $2\pi n_t$ and $2\pi n_b$ after one circle along any closed path. Here n_t and n_b are the integer-valued winding numbers labeling the vortex excitations of the θ_t and θ_b fields. Denoting the winding numbers of the θ_+ and θ_- vortices as n_+ and n_- , the Eq. (7) in the main text yields the following relation

$$n_+ = \frac{n_t + n_b}{2}, \quad n_- = \frac{n_t - n_b}{2} \quad (\text{R1})$$

This relation suggests that, while n_+ and n_- each can be either integer or half integer, they should simultaneously be integers or simultaneously be half integers. Bearing this relation between them, the θ_+ and θ_- fields are no longer decoupled with each other. For example, if the θ_+ field takes an integer-vortex excitation, so does the θ_- field; if instead the θ_+ field takes a half-integer vortex, then the θ_- field will also take a half-integer vortex. This is the ‘‘kinematic constraint’’ between the θ_+ and θ_- fields.

Similar kinematic constraint has been proposed in previous studies (Ref. [10, 17, 18]). In the revised manuscript, we have presented the physical origins of the kinematic constraint between θ_+ and θ_- in more details.

Comment: (ii) There is a statement in the paper: ‘‘As shown in Fig. 2, the kinematic correlation between θ_+ and θ_- has already made the vestigial

phase regimes largely shrink, their extra dynamic coupling might make them further shrink or even vanish". I think the authors should elaborate on this statement. Namely, did you compute this phase diagram both with and without the constraint? (Is that the content of Fig. S3?) Or is there some other way to understand your statement?

Response: We thank the Reviewer very much for the comment! This comment provides us with the opportunity to enhance the quality of our presentation. This statement is elaborated in the following.

Firstly, let us provide the phase diagram with the “kinematic constraint”. Similar with previous studies in Ref. [10, 17, 18], the phase transitions here are determined by the proliferation of the vortices of the θ_+ and θ_- fields. Under the “kinematic constraint” implied by the above Eq. (R1), there exist three types of vortices, i.e. the θ_+ vortices, the θ_- vortices and the half θ_+ -half θ_- vortices. The proliferation of these vortices in different parameter regimes leads to the RG phase diagram shown in Fig. R1(a). In Fig. R1 (a), when the temperature T is low enough, no vortices are excited, and the system takes the $d+id$ SC state indicated by the pink-colored regime. With the enhancement of T , in the low κ/ρ regime, the θ_- vortices first unilateral proliferate to restore the time-reversal symmetry, leading to the charge- $4e$ SC phase indicated by the green-colored regime; in the high κ/ρ regime, the θ_+ vortices first

unilateral proliferate to kill SC, leading to the chiral metal phase indicated by the yellow-colored regime; in the regime with $\kappa \approx \rho$, the half θ_+ -half θ_- vortices first proliferate to simultaneously restore the time-reversal symmetry and kill SC, leading directly to the normal state indicated by the blue-colored regime.

Fig. R1. (a) The RG phase diagram. (b) The Monte-Carlo phase diagram obtained with $\gamma=0$.

Secondly, let us turn off the “kinematic constraint”. It is not practical to turn off the constraint in the RG calculations, as the single-valuedness of the order parameter fields in the continuous space always dictates the constraint. However, we can attempt that in the Monte-Carlo calculations on the lattice. On the lattice, the allowed types of vortices are determined not by the single-valuedness of the order parameter fields, but by free-energy minimization. In the lattice Hamiltonian Eq. (15), the γ term with $\gamma>0$ imposes the constraint via energetically dictating that the n_t and

n_b in Eq. (R1) are integer-valued. The reason is as follow. For an assumed θ_t or θ_b vortex with non-integer winding number, if we circle around the vortex center along any closed path, we would always encounter with a singular point across which the phase angle θ_t or θ_b jumps by a finite angle, leading to a free energy lift in the order of γ . In the whole 2D plane, all the singular points are connected into a singular string, causing a free energy lift in the order of γL , where L denotes the lattice size. Such a free energy cost cannot be compensated via the free energy gain through the entropy which scales with $\text{Ln}(L)$ for large L . Therefore, $\gamma > 0$ in Eq. (15) energetically requires integer n_t and n_b and thus imposes the “kinematic constraint”. This knowledge informs us that we can effectively turn off the “kinematic constraint” through setting $\gamma = 0$ in Eq. (15). Setting $\gamma = 0$, we obtain the Monte-Carlo phase diagram without the “kinematic constraint”, which is shown in Fig. R1 (b). This phase diagram is very simple, which is divided by two lines into four phases touching at one qua-critical point. The straight line parallel to the x-axis represents the K-T transition, suggesting that the (quasi-) ordering temperature of θ_+ only relies on ρ . The line passing through the coordinate origin represents the Ising transition, suggesting that the ordering temperature of θ_- only relies on κ when fixing A . Comparing Fig. R1 (a) and (b), it is found that while the vestigial phases only emerge above the T_c of the d+id SC for sufficiently small or large κ/ρ regimes in (a), they emerge for nearly all

κ/ρ except an isolate point in (b). This comparison suggests that the “kinematic constraint” shrinks the regime for the vestigial phases.

We have added clarification on this point in the revised manuscript both in the main text and the Supplementary Material. Particularly, in the Supplementary Material, we have provided the above Fig. R1(b) and the detailed derivation to understand the physical picture of it.

Comment: (iii) Relatedly, a more clear description of the statement, i.e that coupling between θ_+ and θ_- disfavors the desired phases –is rather important for the logic of the paper since it is used to contrast this proposal from previous proposals based on chiral TSC in monolayers systems. On this point, I think the paper would benefit from some additional contrast between the proposed twisted models and the previously studied untwisted models for chiral condensates.

Response: Many thanks to the Reviewer for the comment! As shown in Fig. R1(a, b), the comparison of the phase diagrams between the cases with and without the “kinematic constraint” suggests that the coupling between the θ_+ and θ_- disfavors the desired vestigial phases. The physical reason underscores this result is explained in the following paragraph.

In the d+id superconducting ground state of TB-QC, there are two

spontaneous symmetry breaking (SB), i.e. the U(1)-gauge SB (strictly speaking, in 2D, this SB is quasi SB) and the time-reversal (TR) SB. When the temperature arises, the two broken symmetries will be restored, leading to phase transitions. The proliferation of the θ_+ (θ_-) vortices leads to restore of U(1)-gauge (TR) symmetry, bringing about the K-T (Ising) transition. When the two fields are decoupled, their vortices proliferate at different temperatures controlled by different parameters in Eq. (12): the ρ for the θ_+ and the κ and A for the θ_- . Therefore, the two phase transitions generally take place at different temperatures, bring about vestigial phase in between the two transition temperatures. Consequently, as shown in Fig. R1(b), except for an isolate κ/ρ point, the vestigial phase emerges at intermediate temperatures for all the κ/ρ regimes. However, when the “kinematic constraint” turns on, the proliferations of the θ_+ and θ_- vortices are correlated, and it is possible that the two phase transitions take place simultaneously in some parameter regimes. As shown in Fig. R1(a), in the regime with $\kappa \approx \rho$, when the half- θ_+ vortices are excited, the “kinematic constraint” requires the half- θ_- vortices should be excited simultaneously. Consequently, both phase transitions take place simultaneously, leading directly to the normal state. In this regime, there exists no vestigial phase. Therefore, the correlation between the θ_+ and θ_- disfavors the vestigial phases.

In the single-layered d+id superconductors, besides the “kinematic constraint”, there exists additional dynamic correlation between the θ_+ and the θ_- fields. As provided in the Supplementary Material of our revised manuscript, the Ginzburg-Landau theory based analysis yields that the continuous Hamiltonian for the single-layered d+id SC is not as simple as Eq. (12). The TB-QC studied here hosts a high symmetry group. Particularly, it is invariant under the C_{2n} rotation accompanied by a succeeding layer exchange. Such a high symmetry group dictates that seven coefficients out of the nine in Eq. (9) are zero, leading to the simple Hamiltonian Eq. (12). The rotation symmetry of the monolayer system is not as high as that of the TB-QC studied here. Consequently, there can be many nonzero coefficients in Eq. (9). Particularly, such term as $\nabla_{\pm}\theta_+ \cdot \nabla_{\pm}\theta_-$ can appear in the Hamiltonian, reflecting the dynamic correlation between the two fields. As clarified on the above, the coupling between the θ_+ and the θ_- fields disfavors the vestigial phase. Therefore, we have reason to conject that the regime for vestigial phases should shrink in the single-layered d+id superconductors.

We have added clarification of the point in the revised manuscript, including both in the main text and the Supplementary Material (SM). Particularly, we have provided derivation of the G-L theory for continuous Hamiltonian for the single-layered d+id SC in the SM.

Comment 2:

Is the γ parameter in Fig. 2 the same γ used in Eq. (9) and/or (15)? I ask because the γ in Eq. (9) should be zero by symmetry, right? Also, for Fig. 2 I think it would be helpful to state that the RG is performed using Eq. (14) and the Monte Carlo using Eq. (15). If that is indeed correct.

Response: Thanks a lot to the Reviewer for the comment, which helps us to enhance the presentation of our manuscript.

We appreciate that the Reviewer has pointed out our oversight in the parameter names. The parameter γ in Fig. 2 is the same as the γ used in Eq. (15). In the revised manuscript, we have used ω instead of γ in Eq. (9). Moreover, in the revised manuscript, we state that the RG is performed using Eq. (14) and the Monte Carlo using Eq. (15) in order to avoid misunderstanding.

Comment 3:

I think that a sentence or two describing the Chiral metal would be very helpful for the reader. Although the authors do provide references, I think it would make for an easier read to just define what this phase is.

Response: Thanks a lot to the Reviewer for the comment! The chiral metal phase is described as follow.

Firstly, in the chiral metal phase, the superconductivity has been killed. So, the Meissner effect and zero resistivity cannot be detected. Secondly, in this phase, the time-reversal symmetry is spontaneously broken. Such time-reversal symmetry breaking can be verified by the polar Kerr effect. Furthermore, there can be spontaneously generated inner magnetic field in the material, which can be detected by the muon spin resonance experiment.

We have added such description of the chiral metal phase in our revised manuscript.

Comment 4:

(i) Why does the fixed point table only include the g-couplings? Is it assumed that all other couplings (ρ and κ) are zero or infinity or something else?

(ii) On the fixed points—which of these are stable/unstable? It would be interesting to add this detail. For example, Eq. (5) and onwards of the work [Park, Ye and Balents Phys. Rev. B 104, 035142 (2021)], outlines a pretty clear method to compute stability of fixed points.

Response: Many thanks to the Reviewer for the professional questions!

The answers to the two questions are as follow.

(i) In the previous manuscript, we had not listed the renormalized values of the stiffness parameters (ρ' and κ'), because they are consistent with the phase revealed by the RG flow result of the g-couplings. Specifically, the ρ' flows to a finite positive value if the U(1)-gauge symmetry is (quasi) broken, otherwise it flows to zero; the κ' flows to infinity if the time- reverse symmetry is broken, otherwise it flows to zero. In addition, although five possible flow results for the normal state are listed in the Table I in the main text, only the first one actually appears in our calculations. We list here the final fixed-point result in Table R1.

Table R1. Fixed points of the coupling parameters under RG, and the corresponding phases

$g_{2,0}$	$g_{0,2}$	g_4	$g_{1,1}$	ρ'	κ'	phase
∞	∞	0	∞	0	0	normal
0	∞	0	0	$> 8/\pi$	0	charge 4e SC
0	0	∞	0	$> 2/\pi$	∞	chiral SC
∞	0	∞	0	0	∞	chiral metal

We have added clarification of this point in the main text and the Supplementary Material.

(ii) We have cited the paper by Park, Ye and Balents. All the fixed points presented here are stable. The analysis of the stability of the fixed points has been added in the Supplementary Material. We outline the standard method here. Firstly, we obtain the fixed points from the RG flow equations. Secondly, we get the differential matrix at each fixed point. Thirdly, all the eigenvalues can be obtained by diagonalizing the differential matrix. If the real parts of all the eigenvalues are negative or

zero at worst, the corresponding fixed point is stable. If the real parts of some eigenvalues are positive, the corresponding fixed point is unstable.

Comment 5:

There are a few typos throughout the main text that should be addressed. Also, in the Supplement, I suspect Eq. (9) has a typo: should it contain a v_{ij} factor instead of one of the γ_{ij} ?

Response: Thanks to the Reviewer for the comments. We appreciate the Reviewer's correction about the spelling problem. In the revised manuscript, we have checked and revised the spelling of the text, which will improve the quality of our paper. As the Reviewer suspected, Eq. (9) in Supplementary Material really has a typo, which has now been corrected. We thank again for the Reviewer's correction!

Summary:

The paper is exciting, thorough and well argued. To improve, I think it would be beneficial to provide a clearer differentiation between the realization of charge-4e superconductivity in the twisted bilayer quasicrystal scheme and the existing proposals in monolayers. Perhaps addressing 'Comment 1' mentioned above would already suffice in achieving this clarity.

Response: Many thanks to the Reviewer for the comments! We

appreciate the high assessment from the Reviewer. Based on the Reviewer's suggestion in **comment 1**, we have supplemented more content about the difference between the realization of charge-4e superconductivity in the TB-QC scheme and the existing proposals in monolayers. We believe that the quality of the paper will be improved by this comparison.

Reviewers' Comments:

Reviewer #1:

Remarks to the Author:

The authors have successfully addressed my comments. Therefore, I recommend its publication.

Reviewer #2:

Remarks to the Author:

I have read the authors' responses to the reviewers, the revised manuscript and supplemental information. The authors have put substantial effort into addressing the comments and revising the manuscript. My concerns have been addressed largely satisfactorily and the revised manuscript is significantly improved over several important aspects. I therefore recommend publication of the revised manuscript in Nature Communications.

Reviewer #3:

Remarks to the Author:

I am satisfied with the authors' responses to scientific criticisms, and I can happily recommend the publication of the manuscript in its present form.